# Comparison of antibiotic prescribing records in two UK primary care electronic health record systems: cohort study using CPRD GOLD and CPRD Aurum databases

Martin C Gulliford [iD],[1] Xiaohui Sun [iD],[1] Thamina Anjuman,[2] Eleanor Yelland,[2] Tarita Murray-Thomas[2]

[1]School of Population Health and Environmental Sciences, King's College London, London, UK
[2]Clinical Practice Research Datalink, Medicines and Healthcare Products Regulatory Agency, London, UK

**Correspondence to**
Dr Martin C Gulliford;
martin.gulliford@kcl.ac.uk

## ABSTRACT

**Objectives** We aimed to evaluate recording of antibiotic prescribing from two primary care electronic health record systems.

**Design** Cohort study.

**Setting** UK general practices contributing to the Clinical Practice Research Datalink (CPRD) databases: CPRD GOLD (Vision data) and CPRD Aurum (EMIS data). English CPRD GOLD general practices were analysed as a subgroup, as all CPRD Aurum practices were located in England.

**Participants** 158 305 patients were randomly sampled from CPRD Aurum and 160 394 from CPRD GOLD.

**Outcome measures** Antibiotic prescriptions in 2017 were identified. Age-standardised and sex-standardised antibiotic prescribing rates per 1000 person years were calculated. Prescribing of individual antibiotic products and associated medical diagnoses was evaluated.

**Results** There were 101 360 antibiotic prescriptions at 883 CPRD Aurum practices and 112 931 prescriptions at 290 CPRD GOLD practices, including 112 general practices in England. The age-standardised and sex-standardised antibiotic prescribing rate in 2017 was 512.6 (95% CI 510.4 to 514.9) per 1000 person years in CPRD Aurum and 584.3 (582.1 to 586.5) per 1000 person years in CPRD GOLD (505.2 (501.6 to 508.9) per 1000 person years if restricted to practices in England). The 25 most frequently prescribed antibiotic products were similar in both databases. One or more medical codes were recorded on the same date as an antibiotic prescription for 72 989 (74%) prescriptions in CPRD Aurum, 84 756 (78%) in CPRD GOLD and 28 471 (78%) for CPRD GOLD in England. Skin, respiratory and genitourinary tract infections were recorded for 39 035 (40%) prescriptions in CPRD Aurum, 41 326 (38%) in CPRD GOLD, with 15 481 (42%) in English CPRD GOLD practices only.

**Conclusion** Estimates for antibiotic prescribing and infection recording were broadly similar in both databases suggesting similar recording across EMIS and Vision systems. Future research on antimicrobial stewardship can also be conducted using primary care data in CPRD Aurum.

## Strengths and limitations of this study

► The study drew on two databases that include large nationwide samples of general practice registered populations.
► Clinical Practice Research Datalink (CPRD) GOLD has UK-wide coverage and comprises data recorded using the Vision practice management system only. At the time of this study, CPRD Aurum general practices were located in England only and included data recorded using the EMIS practice management system. This may contribute to differences between databases.
► We excluded general practices that migrated from Vision to EMIS software system during the study period.
► We employed consistent data definitions and analysis methods across the two databases.
► We analysed data for a single 12-month period; it is possible that changes over time may differ between the two data sources.
► The study only investigated antibiotic prescribing, and related clinical coding; further studies are needed to address other topics of clinical and public health concern.

## INTRODUCTION

Primary care electronic health records (EHRs) provide an important source of longitudinal population-based data for public health research and population health surveillance.[1] In the UK, several EHR databases collect deidentified data from general practices, which are responsible for providing a broad range of general medical services in the primary care setting. The Clinical Practice Research Datalink (CPRD) GOLD database[1] and The Health Improvement Network (THIN) database[2] collect data from general practices that use the Vision practice system; data from EMIS practice systems have been

historically collected by the QResearch database.[3] CPRD has now established a new database, CPRD Aurum[4] that also collects data from general practices using the EMIS practice system. In recent years, there has been a trend for general practices to switch from the Vision practice system to EMIS, with the latter increasing its market share. This has made evaluations of the quality of EMIS data increasingly important for research.

While the research community has nearly 30 years of experience of using Vision data from CPRD GOLD, with numerous studies reporting on data quality,[5 6] less is known about the similarities and differences with data collected in the CPRD Aurum database. EHR systems may offer end users a variety of options, and differing incentives, to code clinical data and record test results. It is therefore possible that analysis of Vision-derived data from CPRD GOLD and EMIS-derived data from CPRD Aurum may yield different findings when substantive research questions are addressed. However, the magnitude of any possible differences between these data sources is largely unexplored. This study aimed to compare results obtained from an analysis of CPRD GOLD and CPRD Aurum data for one exemplar, the recording of antibiotic prescribing.

Antibiotic prescribing in primary care has been the subject of increasing interest in recent years because of the growing awareness that unnecessary prescription of antibiotics in primary care is contributing to the problem of antimicrobial resistance.[7 8] Research using CPRD GOLD[9 10] and THIN[11] showed that the antibiotic prescribing rate is between 500 and 600 antibiotic prescriptions per 1000 patient years, with higher rates at the extremes of age. Nearly half of antibiotic prescriptions may be issued without a clear indication being recorded.[9 11] The present analysis aimed to determine whether analysis of data from CPRD Aurum and CPRD GOLD provides similar estimates with respect to antibiotic prescription and recording.

## METHODS
### Data and participants
The CPRD GOLD database collects data from the four countries of the UK, with 30% of contributing practices located in England at the time of this study. The CPRD GOLD database has been well described[1] and the high quality of the data collected has been documented in many studies.[12] The October 2019 database release from which the study cohort was sampled for this analysis included data on 17.6 million patients, of whom 2.6 million were currently active. In this release, there were 320 currently contributing general practices including 30% in England, 3% in Northern Ireland, 37% in Scotland and 30% in Wales. The CPRD Aurum database is more recently established, and at the time of this study (June 2019 release) drew on data collected from general practices in England only, using the EMIS practice system.[4] The CPRD Aurum database included data on 883 general practices, from which patients were sampled, with 23.1 million patients, including 2.5 million currently active patients. The study required analysis of anonymised data.

A sample of 158 305 patients in CPRD Aurum was taken by randomly selecting 'n' patients from each stratum of general practice, gender and age group. The value of n=9 was selected to provide an appropriate total sample size of just over 150 000. This sampling approach ensured that each general practice was equally represented in the analysis and that age-specific rates would be estimated with equal precision. Age was calculated as the difference between year of birth and 2017. Age groups were categorised as 0–4, 5–14, 15–24, 25–34, 35–44, 45–54, 55–64, 65–74, 75–84 and 85 years and over.

A comparison cohort of patients was extracted from the October 2019 release of CPRD GOLD using the online interface. In this release, there were 290 general practices contributing data to CPRD GOLD throughout 2017, including 112 in England. A sample of 160 394 patients was taken by randomly selecting n=30 patients from each stratum of general practice, gender and age group. Patients were required to have at least 12 months of follow-up in the database estimated as the difference between the latest of their registration start date and 1 January 2017, and the earliest of registration end, practice last collection date, CPRD derived death date and 31 December 2017. General practices that migrated from Vision to EMIS practice systems during 2017 were excluded.

### Measures
We evaluated antibiotic prescribing for the year 2017 because this was the most recent complete year that we included in a larger study of antibiotic prescribing that we report elsewhere.[13] We identified all antibiotic prescriptions issued in 2017, including all drugs from section 5.1 of the British National Formulary (BNF)[14] except antituberculous, antilepromatous agents and methenamine. The BNF includes the following categories of antibiotics: penicillins, cephalosporins (including carbapenems), tetracyclines, aminoglycosides, macrolides, clindamycin, sulfonamides (including combinations with trimethoprim), metronidazole and tinidazole, quinolones, drugs for urinary tract infection (nitrofurantoin) and other antibiotics.

For CPRD GOLD, we employed a list of 2627 antibiotic drugs that were identified from searches of the CPRD GOLD product dictionary browser made by all authors. Searches were made on the drug substance name, product name, BNF chapter and BNF codes. To identify the corresponding products in CPRD Aurum, dm +d codes (the prescribing codes from the National Health Service dictionary of medicines and devices) associated with individual product codes in the CPRD GOLD dictionary browser were mapped to the corresponding dm +d codes in the CPRD Aurum product dictionary browser. A more complete search of the CPRD Aurum product dictionary browser was additionally undertaken on term, product name and drug substance. We also conducted searches using approximate string matching ('fuzzy matching') to match the CPRD Aurum product name to the CPRD GOLD product name or drug substance name from the CPRD GOLD antibiotic code list. The 'agrep' command was used in the R program,[15] using the

Levenshtein edit distance as a measure of approximateness. The resulting code list was edited manually, resulting in 896 CPRD Aurum product codes. CPRD Aurum product codes are up to 17 characters in length and the 'bit64' package in R was employed for data formatting and management.[16] Although more product codes were identified for the CPRD GOLD database, only 195 CPRD GOLD product codes for antibiotics and 167 CPRD Aurum product codes were recorded during 2017.

We analysed medical codes recorded on the same date as antibiotic prescriptions. Medical diagnoses were identified by searching the CPRD GOLD medical dictionary browser for Read terms and inspecting the associated Read chapter hierarchy. As previously reported, all medical codes were subsequently classified as respiratory infections, genitourinary infections, skin infections, eye infections and 'other codes'.[9] The CPRD Aurum medical dictionary includes Read terms, Read codes and SNOMED codes. In order to use the same codes, lists developed for CPRD GOLD were subsequently mapped to CPRD Aurum by matching on Read codes. Evidence of infections was searched in the patient clinical and referral records in CPRD GOLD and in the observation tables in CPRD Aurum. We evaluated whether or not any medical code was recorded on the same date as an antibiotic prescription. We then classified medical codes into 'respiratory infections', 'skin infections', 'genitourinary infections' and other codes.

### Analysis

Age-specific rates were estimated with 95% CIs from the Poisson distribution. Age-standardised and sex-standardised rates, and associated 95% CIs, were calculated per 1000 person years using the 2013 European Standard Population as reference. Null hypothesis significance testing was not undertaken. Instead, we present estimates and 95% CIs to enable the reader to judge whether differences between the databases may be of clinical or epidemiological importance. Potential differences between databases were evaluated using Bland-Altman plots and 95% CIs[17] CPRD GOLD general practices in England were analysed as a subgroup.

### Patient and public involvement

There was no patient or public involvement in the study including development of the research question, selection of outcome measures, study design, conduct or dissemination of findings.

### RESULTS

#### Comparison of antibiotic prescribing rates

In the CPRD Aurum sample, there were 158 305 participants from 883 general practices with 101 360 antibiotic prescriptions during 2017. In the CPRD GOLD sample, there were 160 394 patients from 290 general practices with 112 931 antibiotic prescriptions during 2017. This included 112 general practices in England and 178 in Scotland, Wales or Northern Ireland. The age-standardised and sex-standardised antibiotic prescribing rate was 512.6 (95% CI 510.4 to 514.9) per 1000 PYs in CPRD Aurum and 584.3 (582.1 to 586.5) per 1000 PYs in CPRD GOLD. The rate for CPRD GOLD practices in England was 505.2 (501.6 to 508.9) per 1000 PYs, which is similar to the rate observed in CPRD Aurum.

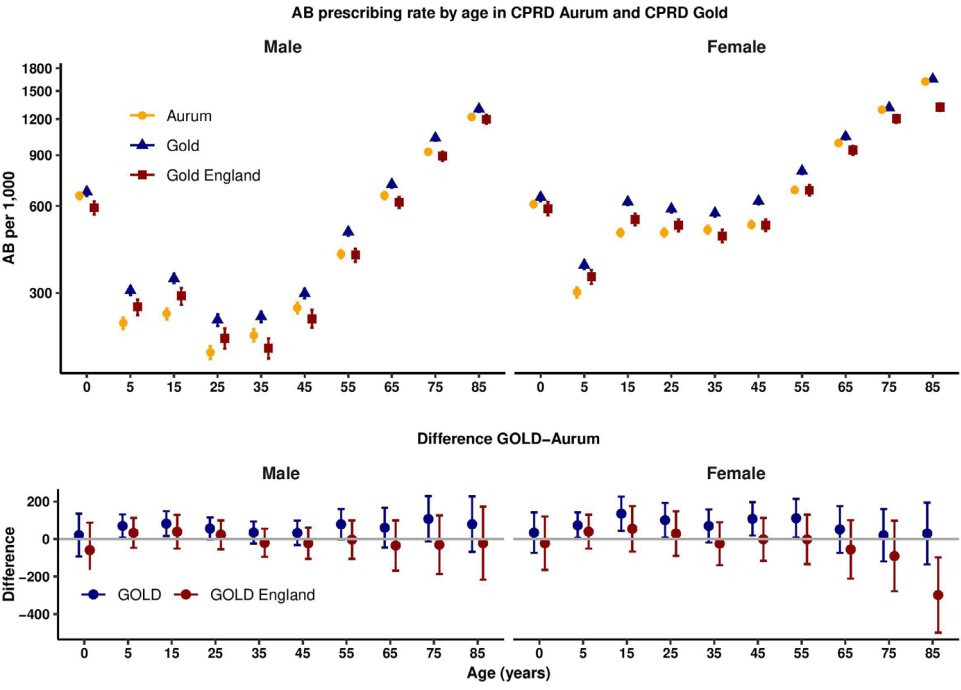

**Figure 1** Antibiotic prescribing rates in Clinical Practice Research Datalink (CPRD) Aurum and CPRD GOLD by age group and sex. Upper figure shows antibiotic prescribing rate per 1000 patient years; lower figure shows difference in antibiotic prescribing rate per 1000 patient years. AB, antibiotics

Figure 1 presents age-specific and sex-specific antibiotic prescribing rates for 2017. Antibiotic prescribing was higher in children under 5 years, decreased until the teenage years, increased again especially in women, before increasing steadily into older ages. This pattern of association was observed in both CPRD Aurum and CPRD GOLD but estimates for CPRD GOLD were slightly higher than for CPRD Aurum, but broadly similar when restricted to CPRD GOLD general practices in England. The lower panel of figure 1 provides a Bland-Altman plot that presents the difference (95% CI) between all CPRD GOLD and CPRD Aurum practices (blue) and CPRD GOLD practices in England only (red). For men and women in all age groups, CPRD GOLD general practices generally had slightly higher antibiotic prescribing rates than CPRD Aurum, while CPRD GOLD general practices in England had broadly similar antibiotic prescribing rates to CPRD Aurum. CIs were compatible with no difference in all except the oldest age group (≥85 years of age) where data are more sparse.

## Most frequently prescribed products

Table 1 presents data for the 25 most frequently prescribed antibiotic products. In CPRD Aurum, amoxicillin 500 mg

**Table 1** Comparison of 25 most commonly prescribed antibiotic products in Clinical Practice Research Datalink (CPRD) Aurum and CPRD GOLD

| Product name | CPRD Aurum | | | CPRD GOLD | | | CPRD GOLD England | | |
|---|---|---|---|---|---|---|---|---|---|
| | Freq. | Per cent | Rank | Freq. | Per cent | Rank | Freq. | Per cent | Rank |
| Amoxicillin 500 mg capsules | 16 684 | 16.5 | 1 | 19 985 | 17.7 | 1 | 6699 | 17.8 | 1 |
| Doxycycline 100 mg capsules | 8993 | 8.9 | 2 | 10 500 | 9.3 | 2 | 2559 | 6.8 | 3 |
| Flucloxacillin 500 mg capsules | 8925 | 8.8 | 3 | 9575 | 8.5 | 3 | 3154 | 8.4 | 2 |
| Trimethoprim 200 mg tablets | 5931 | 5.9 | 4 | 9064 | 8.0 | 4 | 2277 | 6.1 | 4 |
| Nitrofurantoin 100 mg modified-release capsules | 4948 | 4.9 | 5 | 3789 | 3.4 | 7 | 1653 | 4.4 | 6 |
| Clarithromycin 500 mg tablets | 4656 | 4.6 | 6 | 4889 | 4.3 | 5 | 1842 | 4.9 | 5 |
| Phenoxymethylpenicillin 250 mg tablets | 3855 | 3.8 | 7 | 4510 | 4.0 | 6 | 1254 | 3.3 | 7 |
| Amoxicillin 250 mg/5 mL oral suspension sugar free | 3030 | 3.0 | 8 | 3645 | 3.2 | 8 | 1065 | 2.8 | 9 |
| Coamoxiclav 500 mg/125 mg tablets | 2765 | 2.7 | 9 | 2971 | 2.6 | 9 | 1161 | 3.1 | 8 |
| Lymecycline 408 mg capsules | 2379 | 2.4 | 10 | 2916 | 2.6 | 10 | 1010 | 2.7 | 10 |
| Nitrofurantoin 50 mg capsules | 2295 | 2.3 | 11 | 2821 | 2.5 | 11 | 848 | 2.3 | 12 |
| Trimethoprim 100 mg tablets | 2232 | 2.2 | 12 | 2519 | 2.2 | 12 | 912 | 2.4 | 11 |
| Nitrofurantoin 50 mg tablets | 2169 | 2.1 | 13 | 1539 | 1.4 | 18 | 610 | 1.6 | 17 |
| Amoxicillin 250 mg/5 mL oral suspension | 1909 | 1.9 | 14 | 1525 | 1.4 | 19 | 707 | 1.9 | 15 |
| Amoxicillin 125 mg/5 mL oral suspension sugar free | 1802 | 1.8 | 15 | 990 | 0.9 | 25 | 369 | 1.0 | 25 |
| Azithromycin 250 mg tablets | 1602 | 1.6 | 16 | 1306 | 1.2 | 22 | 373 | 1.0 | 24 |
| Erythromycin 250 mg gastroresistant tablets | 1418 | 1.4 | 17 | 1949 | 1.7 | 13 | 727 | 1.9 | 14 |
| Metronidazole 400 mg tablets | 1384 | 1.4 | 18 | 1595 | 1.4 | 17 | 560 | 1.5 | 19 |
| Oxytetracycline 250 mg tablets | 1292 | 1.3 | 19 | 1416 | 1.3 | 20 | 430 | 1.1 | 22 |
| Cefalexin 250 mg capsules | 1241 | 1.2 | 20 | 1661 | 1.5 | 16 | 522 | 1.4 | 20 |
| Ciprofloxacin 500 mg tablets | 1236 | 1.2 | 21 | 1724 | 1.5 | 15 | 662 | 1.8 | 16 |
| Amoxil 500 mg capsules (GlaxoSmithKline UK) | 1193 | 1.2 | 22 | 587 | 0.5 | 33 | 283 | 0.8 | 29 |
| Flucloxacillin 250 mg capsules | 1147 | 1.1 | 23 | 1763 | 1.6 | 14 | 746 | 2.0 | 13 |
| Phenoxymethylpenicillin 125 mg/5 mL oral solution | 1081 | 1.1 | 24 | 712 | 0.6 | 27 | 354 | 0.9 | 26 |
| Amoxicillin 250 mg capsules | 1073 | 1.1 | 25 | 1397 | 1.2 | 21 | 575 | 1.5 | 18 |

**Table 2** Medical coding of antibiotic prescriptions

| | CPRD Aurum | CPRD GOLD | CPRD GOLD England |
|---|---|---|---|
| Number of prescription items | 101 360 | 112 931 | 37 551 |
| Number of prescriptions with unique date | 98 727 | 108 397 | 36 617 |
| Medical code recorded on same date | 72 989 (74.0) | 84 756 (78.2) | 28 471 (77.8) |
| No medical code recorded on same date | 25 738 (26.0) | 23 641 (21.8) | 8146 (22.2) |
| Respiratory infection | 21 350 (21.6) | 26 005 (24.0) | 9549 (26.1) |
| Genitourinary infection | 11 126 (11.3) | 8762 (8.1) | 3315 (9.1) |
| Skin infection | 6559 (6.6) | 6559 (6.1) | 2617 (7.1) |
| Other codes | 33 954 (34.4) | 43 430 (40.1) | 12 990 (35.5) |

Figures are frequencies (per cent of unique prescription dates).
CPRD, Clinical Practice Research Datalink.

capsules, doxycycline 100 mg capsules, flucloxacillin 500 mg capsules, trimethoprim 200 mg tablets and nitrofurantoin 100 mg modified-release capsules represented the five most frequently prescribed products, accounting for 45% of all antibiotic prescriptions. In CPRD GOLD, there were more prescriptions for trimethoprim (8%) and fewer prescriptions for nitrofurantoin (3%), consequently clarithromycin 500 mg tablets and not nitrofurantoin appeared as the fifth ranked product. The same pattern was observed for CPRD GOLD practices in England, although trimethoprim comprised a smaller proportion of all prescriptions than in CPRD GOLD as a whole. Twenty-three of the 25 most frequently prescribed drugs in CPRD Aurum were also in the top 25 ranked prescriptions in CPRD GOLD general practices.

### Recording of medical terms associated with prescriptions
Table 2 summarises data for recording of medical diagnostic codes on the same date as antibiotic prescriptions. Medical codes were recorded on the same date for 72 989 (74%) antibiotic prescriptions in CPRD Aurum, 84 756 (78%) antibiotic prescriptions in CPRD GOLD and 28 471 (78%) for CPRD GOLD in England. Infections of the skin, respiratory tract and genitourinary tracts accounted for 39 035 (40%) of CPRD Aurum prescriptions, 41 326 (38%) of CPRD GOLD prescriptions and 15 481 (42%) of CPRD GOLD prescriptions in England. All other medical codes accounted for 33 954 (34%) in CPRD Aurum, 43 430 (40%) in CPRD GOLD and 12 990 (36%) CPRD GOLD in England.

## DISCUSSION
### Main findings
This analysis shows that antibiotic prescribing estimates from EMIS-derived data in CPRD Aurum are broadly similar to those obtained through analysis of Vision-derived data in CPRD GOLD. This similarity includes the rates of antibiotic prescriptions for subgroups of age and gender, the drug name and strength of antibiotic products prescribed, and the recording of medical diagnoses on the same day as the antibiotic prescription.

We noted that antibiotic was more frequently prescribed in CPRD GOLD than in CPRD Aurum, but this was not the case when the CPRD GOLD sample was restricted to general practices in England. This suggests that antibiotic prescribing may be higher in Wales, Northern Ireland and Scotland where prescription charges have been abolished since 2007, 2010 and 2011, respectively,[18 19] although our study did not investigate the reasons for this difference. As well as slight differences in overall rates, we noted that drug choice might vary between databases. Trimethoprim prescribing was higher in CPRD GOLD than in CPRD Aurum. Nitrofurantoin has been recommended by Public Health England[20] as the drug of first choice for urinary tract infections in adults because of increasing antimicrobial resistance to trimethoprim but this guidance may not apply in the devolved administrations. CPRD GOLD general practices in England were more similar to CPRD Aurum with respect to prescribing of trimethoprim and nitrofurantoin. It is likely that differences in clinical practice between England and the devolved administrations (Scotland, Wales and Northern Ireland) may be greater than the differences between EMIS and Vision practices within England.

Previous primary care database studies have revealed that antibiotic prescriptions are often issued without specific reasons being coded into EHRs. A high proportion of antibiotic prescriptions in the THIN[11] and CPRD GOLD[9] databases are associated with either no code being recorded or non-specific codes being recorded. EHRs systems may offer users discretion over the recording of data items. We found that EMIS data included similar proportions of antibiotic prescriptions being associated with no codes, non-specific codes and codes for infection episodes.

### Strength and limitations
One major strength of our study is that we used real-world data from primary care to estimate rates of antibiotic prescribing. Using data from primary care is likely to provide a reliable picture of prescribing patterns given that about 80% of all antibiotic prescribing in the UK's

National Health Service takes place in primary care.[20] We estimated the difference between CPRD Aurum and CPRD GOLD, as well as the difference between CPRD Aurum and CPRD GOLD in England. A comparison between CPRD GOLD practices in England and CPRD Aurum practices was essential to benchmark recording in the CPRD Aurum database, which at the time of this study comprised contributing practices from England only. We recommend that future comparative studies on antibiotic prescribing in the UK should separately evaluate prescribing in CPRD Aurum with CPRD GOLD practices in England, and CPRD GOLD practices in Wales, Scotland and Northern Ireland, given that factors such as socioeconomic differences, as well as abolished prescription charges, in the devolved nations may have an impact on prescribing in these countries. There were generally only small differences between CPRD Aurum and CPRD GOLD in England. We acknowledge that we could have obtained greater precision with larger samples, but the present approach was pragmatic and provided sufficiently precise estimates for age-specific rates. We did not employ null-hypothesis significance testing but elected to present CIs so that readers could reflect on the substantive importance of any estimated differences for their proposed studies. The community of CPRD researchers collectively has wide experience of compiling code lists for research in the CPRD GOLD database. Less experience is available for the CPRD Aurum database. We noted that CPRD Aurum product codes may be up to 17 characters in length and use of special programming features, such as the bit64 package in R,[16] is required in order to maintain data integrity. We completed extensive searches for product codes in order to identify antibiotic products. We identified a greater number of potential products from the CPRD GOLD data dictionary, but a generally similar number of antibiotic product codes were actually recorded in the two datasets during 2017. Searches in the CPRD Aurum product dictionary should be based on term, drug substance and product names as the BNF classification is less widely available in the CPRD Aurum product dictionary than in CPRD GOLD. It may also be possible to compare 'dm +d' codes from the dictionary of medicines and devices, which are now employed in both CPRD GOLD and CPRD Aurum product dictionaries. We mapped medical code sets between CPRD GOLD and CPRD Aurum by matching on Read codes to make a like-for-like comparison. The analysis shows that, for these conditions, use of the same Read codes gives similar results in CPRD Aurum and CPRD GOLD. There are some medical codes that are only employed in EMIS, which might be omitted through this process, and this merits further evaluation. Experience shows that in Read-coded data, the majority of events are associated with a small number of codes, consequently omission of infrequently used codes is seldom important; our main findings with respect to medical codes were consistent between databases. We also note that records of antibiotic prescribing do not indicate whether medicines were dispensed, whether they were taken or whether they were taken by the patient they were prescribed to or by someone else. It is also possible that prescriptions recorded at out of hours visits, home visits or during attendance at residential care homes may be missing from the patient electronic record. Finally, the analysis undertaken was cross sectional in nature and does not provide evidence about trends in antibiotic recording over time between CPRD GOLD and CPRD Aurum. We used data from July and October 2019 releases of CPRD Aurum and CPRD Gold, respectively; it may be preferable to compare the same month's releases but data for 2017 should be complete by 2019.

## Conclusion

This study finds that analysis of EMIS-derived data in CPRD Aurum gives broadly similar estimates for antibiotic prescribing and infection recording to those reported for Vision-derived data in CPRD GOLD. CPRD GOLD includes general practices in Scotland, Wales and Northern Ireland which have slightly higher antibiotic prescribing than either EMIS or Vision general practices in England. Based on these results, we believe that future research studies can be conducted in CPRD Aurum, informed by previous results from CPRD GOLD or THIN. It may also be possible to combine data from CPRD GOLD English practices with CPRD Aurum data in research on antibiotic prescribing. As CPRD Aurum includes an increasing number of general practices, this database will become increasingly important for public health research. However, further work is needed to better understand the quality and completeness of information recorded in areas such as dosing regimen and treatment duration which are important in estimating treatment exposure in pharmacoepidemiology and pharmacovigilance research.

**Contributors** MCG designed the study; completed the analysis and drafted the paper and is a guarantor. TA sampled CPRD Aurum data. XS developed code lists for CPRD GOLD. TA, EY and TM-T checked and revised code lists for CPRD Aurum. All authors reviewed and contributed to the final draft.

**Funding** The study is funded by the National Institute for Health Research (NIHR) Health Services and Delivery Programme (16/116/46). MCG was supported by the NIHR Biomedical Research Centre at Guy's and St Thomas' Hospitals.

**Disclaimer** The views expressed are those of the authors and not necessarily those of the NHS, the NIHR, or the Department of Health.

**Competing interests** None declared.

**Patient consent for publication** Not required.

**Ethics approval** The protocol for the study received scientific and ethical approval from the CPRD Independent Scientific Advisory Committee (ISAC Protocol 19_110R).

**Provenance and peer review** Not commissioned; externally peer reviewed.

**Data availability statement** Data are available upon reasonable request. Requests for access to data from the study should be addressed to martin.gulliford@kcl.ac.uk. All proposals requesting data access will need to specify planned uses with approval of the study team and CPRD before data release.

**ORCID iDs**
Martin C Gulliford http://orcid.org/0000-0003-1898-9075
Xiaohui Sun http://orcid.org/0000-0003-1576-4903

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
