## [Reviewer comments · BMJ Open]

ARTICLE DETAILS

TITLE (PROVISIONAL)	Comparison of antibiotic prescribing records in two UK primary care electronic health record systems: Cohort study using CPRD GOLD and CPRD Aurum databases
AUTHORS	Gulliford, Martin; Sun, Xiaohui; Anjuman, Thamina; Yelland, Eleanor; Murray-Thomas, Tarita

VERSION 1 – REVIEW

REVIEWER	Dr Kate Honeyford Imperial College, London, UK
REVIEW RETURNED	31-Mar-2020

GENERAL COMMENTS	This is an important paper, which is easy to read and covers an important topic. Although I have said major revisions, I think it is possible to address them relatively quickly. Main points A. CPRD GOLD England Clearly the distinction with CPRD GOLD and why this would be different and needs to be considered when comparing with CPRD Aurum is really important. As an English researcher with background knowledge I can infer why you needed to separate these out, but this needs to be made more explicit for an international audience. In addition, it is not clear why you sampled from CPRD GOLD, and then looked at a sub-sample of English practices, and not create three samples CPRD GOLD NOT ENGLAND, CPRD ENGLAND and CPRD Aurum. You set out to sample approximately 150,000 patients from both CPRD Aurum and CPRD GOLD, but you then compare CPRD GOLD ENGLAND, which is a much smaller sample. B. Statistical analysis/quantification It is a descriptive study and that is fine, but throughout it is hard to determine the weight of phrases such as 'similar' without any statistical tests. For example, in the last sentence of the first paragraph of the results – 505.2 (501.6-508.9) cf 512.6 (510.4-514.9) – are described as 'similar'. The CIs don't overlap – how are you defining similar? This is a challenge for me throughout the paper, results and discussion. In Figure 1 you have shown age-specific rates with error bars, because the points are 'off' so they can be seen, it is not possible to see if the error bars overlap, and in the text there is no description of the Bland-Altman plots – can you help the reader interpret them? The only age-gender rate where GOLD ENGLAND is different to Aurum is age 85+ female – do you think this is important? Later you say 'there were more prescriptions for trimethoprim' – can you quantify this? – is it important?
--

	Is 74% compared to 78% for recording of medical codes important? C. Recording of medical diagnostic codes This seems really important and is discussed, with recommendations in the discussion, but is quite technical. I wonder whether this could be summarised in a flow diagram, the section in the methods was difficult to follow. D. Strengths and Weaknesses I am not convinced that the S&W relate to the objective. You found some differences in recording but you argue that they are similar and therefore are not that important. Although there are difference outside England, which your study was not designed to investigate, so it may be difficult to conclude this.
--	--

REVIEWER	Laura Shallcross University College London, Institute of Health Informatics
REVIEW RETURNED	14-Apr-2020

GENERAL COMMENTS	Summary This study compares recording of antibiotic prescribing in two large primary care databases. This is important, since historically most research has been conducted in CPRD GOLD, but it is likely that researchers will increasingly start using CPRD Aurum. The study has a simple, clear study design and the results are useful for anyone who plans to use these datasets to study antibiotic prescribing. Major comments  • The authors compare CPRD Aurum to the subset of English CPRD GOLD practices. However, I could not find a breakdown of the number of CPRD-GOLD practices for each of the four nations. This should be included. • The number of practices included in CPRD-GOLD has decreased as practices shift from Vision software to EMIS. This could limit the generalisability of research undertaken using CPRD-GOLD, and makes the case for using Aurum or a combination of Aurum and GOLD. A comment on the trend for practices to switch to EMIS should be included. Specific comments Abstract: Line 29 – please also include the number of CPRD GOLD practices that were in England. Line 35-36 “one or more medical code” – might be helpful to make it clear that this includes infective and non-infective codes.... Conclusion: worth stating that a benefit of Aurum is that it includes a larger number of practices? Strengths and limitations “We excluded general practices that migrated between software systems” – were practices that migrated at any point excluded, or only those that migrated during the period of data analysis ie 2017? Introduction: Lines 30/31. Worth also making the point that the number of practices using Vision software is declining, making it important to capture/analyse data from practices that use EMIS. Methods Data and participants: Lines 32/33 “CPRD GOLD: 17.6 million patients, 2.6 million currently active. CPRD Aurum: 883 practices,
---

	23.1 million patients, 2.5 million active.” Please list the number of practices in CPRD GOLD in this section. Please make it clear that you sampled from all 883 CPRD Aurum practices. Measures – page 6 Line 40: “We evaluated antibiotic prescribing for the year 2017” - was this total number of prescriptions for any of the listed drugs? Results Line 40: Please add in the number of CPRD GOLD practices in England, Wales, Scotland and Northern Ireland. Discussion Line 40/41 “implies that antibiotic prescribing may be higher in Wales, Scotland, NI where there are no prescription charges” – is this the most likely explanation? What about increased levels of ill-health in these nations? Less than 10% of the population in England pay prescription charges https://www.kingsfund.org.uk/sites/default/files/field/field_publication_file/Commission%20Final%20%20interactive.pdf page 25. Lines 55-58: Agree – differences in the 2 datasets are most likely to be driven by variation in prescribing practices across the 4 nations. Are there differences in recommended first line therapy for UTI in the 4 nations? I think Scottish guidance for lower UTI recommends either trimethoprim or nitrofurantoin, as opposed to England where nitrofurantoin is preferred. https://www.sign.ac.uk/assets/sign88.pdf Over time the number of practices using Vision software has declined, with more practices using EMIS. This is a strong incentive for researchers to start analysing data from CPRD Aurum rather than CPRD GOLD (or to use both). It would be useful for the authors to comment on this in their discussion, or to include a figure) on the number of practices using Vision or EMIS software over time. Figure 1. Are the units for “difference” (lower graph) also antibiotic prescriptions per 1000 individuals?
--	---

VERSION 1 – AUTHOR RESPONSE

Reviewer: 1

This is an important paper, which is easy to read and covers an important topic.

Thank you for this feedback.

A. CPRD GOLD England

Clearly the distinction with CPRD GOLD and why this would be different and needs to be considered when comparing with CPRD Aurum is really important. As an English researcher with background knowledge I can infer why you needed to separate these out, but this needs to be made more explicit for an international audience. In addition, it is not clear why you sampled from CPRD GOLD, and then looked at a sub-sample of English practices, and not create three samples CPRD GOLD NOT ENGLAND, CPRD ENGLAND and CPRD Aurum. You set out to sample approximately 150,000 patients from both CPRD Aurum and CPRD GOLD, but you then compare CPRD GOLD ENGLAND,

which is a much smaller sample.

Thank you for this point. At the time of this study CPRD Aurum included data from contributing practices in England only. An analysis of CPRD GOLD limited to English practices was therefore essential to benchmark recording in the CPRD Aurum database. Furthermore, as researchers routinely use CPRD GOLD to achieve greater representativeness of the UK population we opted for an analysis comparing CPRD GOLD, CPRD GOLD England and CPRD Aurum to also provide insights on recording in the CPRD GOLD database as a whole. We acknowledge that undertaking the analysis as suggested may have provided clearer insights on the variation in antibiotic prescribing across countries but this was not a primary objective of our analysis.

We now explain under limitations (p12): 'A comparison between CPRD GOLD practices in England and CPRD Aurum practices was essential to benchmark recording in the CPRD Aurum database, which at the time of this study comprised of contributing practices from England only. We recommend that future comparative studies on antibiotic prescribing in the UK should separately evaluate prescribing in CPRD Aurum with CPRD GOLD practices in England, and CPRD GOLD practices in Wales, Scotland and Northern Ireland, given that factors such as socioeconomic differences, as well as abolished prescription charges, in the devolved nations may have an impact on prescribing in these countries.'

B. Statistical analysis/quantification

It is a descriptive study and that is fine, but throughout it is hard to determine the weight of phrases such as 'similar' without any statistical tests. For example, in the last sentence of the first paragraph of the results – 505.2 (501.6-508.9) cf 512.6 (510.4-514.9) – are described as 'similar'. The CIs don't overlap – how are you defining similar? This is a challenge for me throughout the paper, results and discussion. In Figure 1 you have shown age-specific rates with error bars, because the points are 'off' so they can be seen, it is not possible to see if the error bars overlap, and in the text there is no description of the Bland-Altman plots – can you help the reader interpret them? The only age-gender rate where GOLD ENGLAND is different to Aurum is age 85+ female – do you think this is important? Later you say 'there were more prescriptions for trimethoprim' – can you quantify this? – is it important? Is 74% compared to 78% for recording of medical codes important?

Thank you for these comments. We now explain:

Methods (p8): 'Null hypothesis significance testing was not undertaken. Instead, we presented estimates and 95% confidence intervals to enable the reader to judge whether differences between the databases may be of clinical or epidemiological importance.'

Results (p9): 'The lower panel of Figure 1 provides a Bland-Altman plot that presents the difference (95% confidence interval) between CPRD GOLD and CPRD Aurum for all CPRD GOLD practices (blue) and CPRD GOLD practices in England (red)... Confidence intervals were compatible with no difference in all except the oldest age-group, where data are more sparse.'

And 'In CPRD GOLD, there were more prescriptions for trimethoprim (8%) and fewer prescriptions for nitrofurantoin (3%)'

Discussion (p12): 'We did not employ null-hypothesis significance testing but elected to present confidence intervals so that readers could reflect on the substantive importance of any estimated differences for their proposed studies.'

C. Recording of medical diagnostic codes

This seems really important and is discussed, with recommendations in the discussion, but is quite

technical. I wonder whether this could be summarised in a flow diagram, the section in the methods was difficult to follow.

Thank you, we now provide additional clarification (p8): 'We evaluated whether or not any medical code was recorded on the same date as an antibiotic prescription. We then classified medical codes into 'respiratory infections', 'skin infections', 'genito-urinary infections', and 'other codes'.'

D. Strengths and Weaknesses

I am not convinced that the S&W relate to the objective. You found some differences in recording but you argue that they are similar and therefore are not that important. Although there are difference outside England, which your study was not designed to investigate, so it may be difficult to conclude this.

Thank you for this comment. We now add an additional bullet point (page 3): '• CPRD GOLD has UK-wide coverage and comprises data recorded using the Vision practice management system only. At the time of this study, CPRD Aurum general practices were located in England-only and included data recorded using the EMIS practice management system. This may contribute to differences between databases.'

Reviewer: 2

Reviewer Name: Laura Shallcross

Summary

The study has a simple, clear study design and the results are useful for anyone who plans to use these datasets to study antibiotic prescribing.

Thank you for this feedback.

Major comments

- The authors compare CPRD Aurum to the subset of English CPRD GOLD practices. However, I could not find a breakdown of the number of CPRD-GOLD practices for each of the four nations. This should be included.*

Thank you, we now explain (p5): 'In this release, there were 320 currently contributing general practices including 30% in England, 3% in Northern Ireland, 37% in Scotland and 30% in Wales.'

- The number of practices included in CPRD-GOLD has decreased as practices shift from Vision software to EMIS. This could limit the generalisability of research undertaken using CPRD-GOLD, and makes the case for using Aurum or a combination of Aurum and GOLD. A comment on the trend for practices to switch to EMIS should be included.*

Thank you, we now add (p4): 'In recent years, there has been a trend for general practices to switch from the Vision practice system to EMIS, with the latter increasing its market share. This has made evaluations of the quality of EMIS data increasingly important for research.'

Specific comments

Abstract:

Line 29 – please also include the number of CPRD GOLD practices that were in England.

Thank you, this has been added (p2).

Line 35-36 “one or more medical code” – might be helpful to make it clear that this includes infective and non-infective codes....

Thank you, this now reads (p2): ‘One or more medical codes were recorded on the same date as an antibiotic prescription.’

Conclusion: worth stating that a benefit of Aurum is that it includes a larger number of practices?

Thank you, we now add (p14): ‘CPRD Aurum includes a greater number of general practices and will become increasingly important if general practices continue to migrate from Vision to the EMIS practice system.’

Strengths and limitations

“We excluded general practices that migrated between software systems” – were practices that migrated at any point excluded, or only those that migrated during the period of data analysis ie 2017?

Thank you, we now say (p3): ‘We excluded general practices that migrated between software systems during the study.’

Introduction: Lines 30/31. Worth also making the point that the number of practices using Vision software is declining, making it important to capture/analyse data from practices that use EMIS.

Thank you, we now add on page 4: ‘In recent years, there has been a trend for general practices to switch from the Vision practice system to EMIS, with the latter increasing its market share. This has made evaluations of the quality of EMIS data increasingly important for research.’

Methods

Data and participants: Lines 32/33 “CPRD GOLD: 17.6 million patients, 2.6 million currently active. CPRD Aurum: 883 practices, 23.1 million patients, 2.5 million active.” Please list the number of practices in CPRD GOLD in this section.

Please make it clear that you sampled from all 883 CPRD Aurum practices.

Thank you, we now explain (p5): ‘The CPRD Aurum database included data on 883 general practices, from which patients were sampled’

Measures – page 6

Line 40: “We evaluated antibiotic prescribing for the year 2017” - was this total number of prescriptions for any of the listed drugs?

Thank you, we now explain (p6): ‘We identified all antibiotic prescriptions documented in the patient record in the in 2017, including all drugs from section 5.1 of the British National Formulary (BNF)’

Results

Line 40: Please add in the number of CPRD GOLD practices in England, Wales, Scotland and Northern Ireland.

Thank you, we now say (p9): 'This included 112 general practices in England and 178 in Scotland, Wales or Northern Ireland.'

Discussion

Line 40/41 "implies that antibiotic prescribing may be higher in Wales, Scotland, NI where there are no prescription charges" – is this the most likely explanation? What about increased levels of ill-health in these nations?

Thank you, we now say (p11): 'This suggests that antibiotic prescribing may be higher in Wales, Northern Ireland and Scotland where prescription charges have been abolished since 2007, 2010 and 2011 respectively,(18, 19) although our study did not investigate the reasons for this difference.'

Lines 55-58: Agree – differences in the 2 datasets are most likely to be driven by variation in prescribing practices across the 4 nations.

Are there differences in recommended first line therapy for UTI in the 4 nations? I think Scottish guidance for lower UTI recommends either trimethoprim or nitrofurantoin, as opposed to England where nitrofurantoin is preferred.

Thank you, we now say (p11): 'Nitrofurantoin has been recommended by Public Health England (18) as the drug of first choice for urinary tract infections in adults because of increasing antimicrobial resistance to trimethoprim but this guidance may not apply in the devolved administrations.'

Over time the number of practices using Vision software has declined, with more practices using EMIS. This is a strong incentive for researchers to start analysing data from CPRD Aurum rather than CPRD GOLD (or to use both). It would be useful for the authors to comment on this in their discussion, or to include a figure) on the number of practices using Vision or EMIS software over time.
Thank you, we now add (p14): 'CPRD Aurum includes a greater number of general practices and will become increasingly important for research if general practices continue to migrate from Vision to the EMIS practice system.'

Figure 1. Are the units for "difference" (lower graph) also antibiotic prescriptions per 1000 individuals?

Thank you, we now add to the legend of Figure 1: 'Upper figure shows antibiotic prescribing rate per 1,000 patient years; lower figure shows difference in antibiotic prescribing rate per 1,000 patient years.'

VERSION 2 – REVIEW

REVIEWER	Dr Kate Honeyford Imperial College, London, UK
REVIEW RETURNED	18-May-2020
GENERAL COMMENTS	The clarifications have made it much easier to read. It is a useful and informative study.
REVIEWER	Laura Shallcross University College London, UK
REVIEW RETURNED	25-May-2020
GENERAL COMMENTS	The authors have addressed all my comments and I have nothing further to add.